# Retreatment with Cisplatin May Provide a Survival Advantage for Children with Relapsed/Refractory Hepatoblastoma: An Institutional Experience

**DOI:** 10.3390/cancers15153921

**Published:** 2023-08-01

**Authors:** Katherine M. Somers, Rachel Bernstein Tabbouche, Alexander Bondoc, Alexander J. Towbin, Sarangarajan Ranganathan, Greg Tiao, James I. Geller

**Affiliations:** 1Division of Pediatric Hematology/Oncology, Cincinnati Children’s Hospital Medical Center, Cincinnati, OH 45229, USA; 2Department of Pediatric and Thoracic Surgery, Cincinnati Children’s Hospital Medical Center, Cincinnati, OH 45229, USA; alex.bondoc@cchmc.org (A.B.); greg.tiao@cchmc.org (G.T.); 3Department of Radiology, Cincinnati Children’s Hospital Medical Center, Cincinnati, OH 45229, USA; alexander.towbin@cchmc.org; 4Department of Radiology, University of Cincinnati College of Medicine, Cincinnati, OH 45267, USA; 5Department of Pathology and Laboratory Medicine, Cincinnati Children’s Hospital Medical Center, Cincinnati, OH 45229, USA; sarangarajan.ranganathan@cchmc.org

**Keywords:** hepatoblastoma, relapsed hepatoblastoma, cisplatin, pediatric liver tumors

## Abstract

**Simple Summary:**

Treatment of hepatoblastoma at time of first diagnosis has evolved to include well described chemotherapy regimens along with surgical resection, resulting in improved outcomes. Unfortunately, in cases of recurrent or refractory disease, there is no standard of care to employ and these patients are largely under described in the literature. The aim of this retrospective study was to describe the characteristics, treatment and outcomes of a cohort of patients with relapsed or refractory hepatoblastoma at our institution. We report 50% overall survival, which is consistent with other published data, and found that cisplatin remains the most effective chemotherapy agent with improved outcomes.

**Abstract:**

Background: Hepatoblastoma (HB) is the most common liver malignancy in children. There is no standard of care for management of relapsed/refractory HB (rrHB) and reports in the literature are limited. Objective: To describe presenting features, biology, treatment strategies, and outcomes for pediatric patients with relapsed/refractory hepatoblastoma. Methods: An IRB-approved retrospective institutional review of patients with rrHB who presented for consultation and/or care from 2000–2019. Clinical, radiographic, and histologic data were collected from all patients. Results: Thirty subjects were identified with a median age of 19.5 months (range 3–169 months) at initial diagnosis and 32.5 months (range 12–194 months) at time of first relapse. 63% of subjects were male, 70% Caucasian, and 13% were born premature. Three subjects had a known cancer predisposition syndrome. Eight patients had refractory disease while 22 patients had relapsed disease. Average time from initial diagnosis to relapse or progression was 12.5 months. Average alpha-fetoprotein (AFP) at initial diagnosis was 601,203 ng/mL (range 121–2,287,251 ng/mL). Average AFP at relapse was 12,261 ng/mL (range 2.8–201,000 ng/mL). For patients with tumor sequencing (*n* = 17), the most common mutations were in CTNNB1 (13) and NRF2 (4). First relapse sites were lungs (*n* = 12), liver (*n* = 11) and both (*n* = 6). More than one relapse/progression occurred in 47% of subjects; 6 had ≥3 relapses. Pathology in patients with multiply relapsed disease was less differentiated including descriptions of small cell undifferentiated (*n* = 3), pleomorphic (*n* = 1), transitional liver cell tumor (*n* = 2) and HB with carcinoma features (*n* = 1). All subjects underwent surgical resection of site of relapsed disease with 7 subjects requiring liver transplantation. Overall survival was 50%. Survival was associated with use of cisplatin at relapse (78.6% with vs. 25% without, *p* = 0.012). The most common late effect was ototoxicity with at least mild sensorineural hearing loss found in 80% of subjects; 54% required hearing aids. Conclusions: Retreatment with cisplatin at the time of relapse may provide an advantage for some patients with hepatoblastoma. Multiply relapsed disease was not uncommon and not associated with a worse prognosis. Careful attention should be paid to cumulative therapy-induced toxicity while concurrently aiming to improve cure.

## 1. Introduction

Hepatoblastoma (HB) is the most common form of liver cancer in children, accounting for approximately 1% of pediatric cancers. The prognosis for upfront therapy has improved with a combined therapeutic approach including surgical resection and chemotherapy [1,2]. Approximately 10–15% of HB patients who achieve a complete remission (CR) experience a relapse [3], and 10–30% of patients with metastatic HB never achieve a remission with initial upfront therapy [4], and develop refractory disease. Presenting features, treatment, and outcomes of relapsed HB have been insufficiently described. To date, there are only two relatively large case series published centered on describing experiences with relapsed hepatoblastoma, one from the SIOPEL1-3 studies in Europe (59 patients) and a second from the INT0098 study in the United States (36 patients) [3,5]. Similar to relapsed HB, patients with refractory disease who failed to achieve a CR during upfront therapy are underrepresented in the literature. 

In addition to a lack of published data, advancing standards for relapsed HB therapy is confounded by the inherent heterogeneity of the rrHB patient population with regards to disease burden, prior therapy intensity, underlying comorbidity, toxicities from prior therapy and other factors such as access to liver transplantation. Prior therapy intensity is largely driven by defined risk stratification groups present at first diagnosis which consider Pretreatment Extent of Tumor (PRETEXT) group and associated annotation factors (multifocality, rupture, approximation to veins, extrahepatic extension), tumor histology, serum alpha-fetoprotein (AFP) levels, age, and metastases [6]. Serum alpha-fetoprotein levels can also be used as markers of disease response and predictor of outcome in many, but not all, patients [7]. First-line standards of care account for such various prognostic criteria resulting in risk-adapted therapy [6]. Varying co-morbidities impacting therapy at all phases of care include inherent organ dysfunction that may result from co-morbidities related to prematurity, cancer predisposition syndromes such as Beckwith Wiedeman Syndrome and Trisomy 18, or co-morbid congenital organ dysfunction such as renal insufficiency. 

Platinum-based agents have been at the core of improvement in overall survival rates for upfront therapy in multiple international trials [2,6,8,9,10]. Current front-line therapy intensity ultimately ranges from observation without chemotherapy following upfront surgery (very rare), to therapy including a combination of surgery and cisplatin with or without doxorubicin (and with/without additional agents such as vincristine and 5-fluorouracil) ranging from 2 doses of single agent cisplatin (PHITT/AHEP1531, NCT03533582) to 6 cycles of 4-drug therapy, to interval compressed cisplatin-based therapy including doxorubicin and other agents. Liver transplant is indicated for approximately 15–20% of HB patients [11]. Such chemotherapy regimens combined with current surgical options are often effective but carry risk of toxicity, notably permanent ototoxicity and mostly reversible renal toxicity from cisplatin, cardiotoxicity and secondary cancers from doxorubicin, and co-morbidities related to possible liver transplantation [6,7]. 

While international multi-cooperative group trials such as PHITT/AHEP1531 have been advanced for newly diagnosed patients, there is no widely agreed upon consensus on the most appropriate chemotherapy regimen and surgical interventions for management of relapsed or progressive hepatoblastoma, and such patients have not been the focus of formal cooperative group study [5,12,13,14,15]. Resulting variability of intensity of front-line therapy presents corresponding variability in patients who relapse, with more gently treated patients possibly having more options in the second-line setting. In this study, the initial and relapse presentation, treatment courses, responses and outcomes of another relatively large cohort of patients with rrHB is presented, wherein cisplatin retreatment is identified as potentially advantageous in the management of relapsed HB. 

## 2. Methods

Following IRB approval, a retrospective review of the medical records of pediatric patients (less than 18 years old) who presented to CCHMC for clinical consultation and/or care of rrHB from 1 January 2000 to 1 July 2019 was undertaken. No patients that received care at CCHMC during this time frame for rrHB were excluded. Subjects were identified utilizing the Cincinnati Children’s cancer database. Clinical data extracted from the medical record included: demographics, presenting features, treatments, toxicities, and outcomes. 

### 2.1. Pathology Review

An independent pathology review including immunohistochemical staining of all samples from time of initial diagnosis and relapse was performed by the CCHMC Department of Pathology for clinical purposes. This data was extracted from the medical record for analysis in this study.

### 2.2. Genetic Analysis

Next-generation sequencing was obtained on formalin-fixed, paraffin embedded tumor samples, when available, at initial presentation and relapse through FoundationOne™ (Roche, Cambridge, MA, USA) to assess mutational burden and clinically actionable genetic variants.

### 2.3. Evaluation of Response

Tumor response was assessed by change in alpha-fetoprotein (AFP) and changes identified on imaging. A board-certified pediatric radiologist with more than 10 years of experience reviewed the imaging components of this study for clinical care purposes. Computed tomography scan and/or magnetic resonance imaging obtained at diagnosis were reviewed to determine the PRETEXT score, based on Children’s Oncology Group (COG) modifications to the 2017 PRETEXT guidelines. Evaluation was completed of primary tumor site and pulmonary metastases when applicable.

Using the RECIST criteria, complete response (CR) was defined as complete imaging response with no evidence of disease and normalization of the AFP. Partial response (PR) was defined as >30% reduction of the tumor on imaging and/or a drop in AFP > 1 log fold without growth on imaging. Progressive disease (PD) was defined as >20% growth on imaging or a serial increase in AFP over three consecutive timepoints. Stable disease (SD) occurred when subjects did not meet the definition for response or progression. Multiply relapsed disease was defined as new evidence of disease based on imaging or by AFP when a remission had previously been documented. 

Cisplatin chemosensitivity was defined as >1 log fold decrease in AFP and/or >30% reduction of tumor volume.

### 2.4. Statistical Analysis

Cisplatin survival data was analyzed using a one-sided Fischer’s exact test to evaluate for statistical significance. Due to the limited sample size, descriptive statistical analyses were primarily utilized for this study. Age, gender, time from initial diagnosis to relapse, sites of upfront disease, upfront response to cisplatin, and use of cisplatin at relapse were evaluated for impact on survival. Kaplan-Meier curves were created using GraphPad Prism (San Diego, CA, USA). A log-rank (Mantel-Cox) test was used to calculate *p*-values.

## 3. Results

### 3.1. Demographics

Thirty subjects were identified meeting criteria for inclusion, of which 63% were male (Table 1). Average/median age at initial diagnosis is 28.8 months/19.5 months (range 3–169 months) and 41.3 months/32.5 months at relapse (range 12–194 months). 21 patients identified as non-Hispanic white, 4 as Hispanic, 2 as black, 2 as Middle Eastern and 1 subject as Pacific Islander. Three patients presented with predisposition syndromes (Beckwith-Weidemann, Trisomy 18 and ARPKD/Caroli) and four patients were born premature at a gestational age of less than 37 weeks. 

### 3.2. Initial Presentation and Treatment

60% (18/30) of patients were classified as Evans Stage IV disease with metastases at initial diagnosis, 27% (8) were Evans Stage III, 3% (1) were stage II and 10% (3) were stage I. The most frequent site of metastatic disease was in the lung (*n* = 14), with 3 subjects having extrahepatic metastatic disease in the abdomen. Average alpha-fetoprotein (AFP) at time of initial diagnosis was 601,203 ng/mL (range 121–2,287,251 ng/mL) for subjects in which the initial AFP did not exceed the upper limit of detection for the assay. The most common histology at diagnosis was mixed epithelial (50%), epithelial embryonal (10%), epithelial fetal (10%), and epithelial + other (27%). Of the 8 subjects that were noted to be cisplatin resistant during upfront therapy, histology of upfront disease included mixed fetal/embryonal with small cell undifferentiated INI-1 retained components (2), mixed fetal/embryonal with small cell undifferentiated and rhabdoid features (1), mixed epithelial with embryonal (2), mixed fetal and embryonal with mactrabecular pattern (1) and mixed epithelial with mesenchymal components with (1) and without (1) teratoid features. 

During upfront treatment, this cohort underwent a total of 38 surgical interventions not including diagnostic biopsies. These operations included 19 occurrences of hepatic mass resections, 10 thoracotomies with lung nodule resections, and 9 liver transplantations. 15 subjects had greater than one operation during upfront therapy. 

Subjects were treated with multiple chemotherapy regimens, including cisplatin, 5-floururacil, vincristine and doxorubicin per COG AHEP0731 protocol and cisplatin with doxorubicin per SIOPEL4 A1-3. All 30 patients received at least one dose of cisplatin as a component of their initial treatment course. 18 patients were identified to have cisplatin sensitive disease during upfront therapy while 8 were noted to be cisplatin resistant; 4 received cisplatin as part of adjuvant chemotherapy after upfront resection which itself resulted in a CR. Other commonly used chemotherapy agents during upfront therapy included vincristine (27), 5-fluoruracil (25), doxorubicin (22), carboplatin (12) and irinotecan (10). 

### 3.3. Tumor Genetics

Genetic testing shown in Table 2 was available for 17 patients. 76% of those who had genetic testing harbored a mutation of CTNNB1 and 24% a NRF2 mutation. All 4 patients with a NRF2 mutation experienced multiple relapses. CTNNB1 in conjunction with MDM4 mutation was observed in 2 patients, both of whom died secondary to their disease. Of the 17 subjects with available tumor genetic testing, using the limited Foundation One genetic platform including over 300 genes, 41% (7) had one abnormality identified, 24% (4) harbored 2 identifiable mutations, 24% (4) had 3 mutations and 12% (2) of subjects were found to have greater than 3 mutations, demonstrating the complexity of genetic alterations in this population.

### 3.4. Relapse Presentation and Treatment

Twenty-two patients experienced disease relapse and 8 were classified as having refractory disease during upfront therapy. 47% (13/30) of patients ultimately experienced multiply relapsed HB, with 6 patients experiencing three or more disease recurrences. 

Time from initial diagnosis to relapse presentation or progression ranged from 3.9 months to 52 months, with a median/average of 10.6/12.5 months. Relapse sites included the lungs (*n* = 12), liver or surgical bed (*n* = 11) and both lung and liver (*n* = 6). Alpha-fetoprotein levels at time of first relapse were normal (*n* = 2), between 10–1000 ng/mL (*n* = 16), greater than 1000 ng/mL (*n* = 9) and not available in 3 subjects. Average AFP at time of first relapse was 12,261 ng/mL with a range of 2.9–201,000 ng/mL. 

In the patients with multiply relapsed disease, when histology was available, pathology at time of relapse included mixed fetal and embryonal (*n* = 4), small cell undifferentiated (3), pleomorphic (1), transitional liver cell tumor (2) and HBL with hepatocellular carcinoma features (1). 

Management strategies at time of first relapse included both chemotherapy and surgery (22), chemotherapy with surgery and radiation therapy (2), surgery alone (2), chemotherapy alone (1), or palliative care without anti-cancer therapy (3). A variety of chemotherapy regimens were used during management of rrHB. Most commonly used regimens included cisplatin with or without doxorubicin, irinotecan, ifosfamide/carboplatin/etoposide, and less commonly included gemcitabine/oxaliplatin, sorafenib/rapamycin, pembrolizumab, ifosfamide/doxorubicin, temsirolimus/temozolamide/irinotecan, pazopanib, and cisplatin with SAHA. At time of first relapse, 29 surgical interventions were undertaken in this cohort including thoracotomy with resection (15), liver transplantation (6), hepatectomy (6), multi-visceral transplantation of liver and kidney (1) and laparotomy with ablation (1). 

The most common late effect at the last point of follow-up following salvage therapy was ototoxicity. Sensorineural hearing loss classified as at least mild was found in 80% of subjects. Of the subjects where data was available, 54% required hearing aids. Of the subjects who were treated with additional cycles of cisplatin at the time of relapse, 7 patients (14 ears) were identified as having sufficient baseline (at time of relapse) and follow-up audiological data points. The average change in hearing from baseline at initial diagnosis to final data points (end of therapy or death) was 16.44 dB and from time of relapse to final (end of therapy or death) was 9.99 dB. 

### 3.5. Outcomes

The overall survival for relapsed refractory patients was 50% (15/30 subjects), with one subject currently receiving ongoing chemotherapy. Subjects who died due to disease had a shorter average time between initial diagnosis and relapse as compared to those who survived (9.11 months vs. 15.45 months, *p* = 0.06). 

Cisplatin was re-administered to 14 patients at time of relapse with 77% (7/9) patients with evaluable disease at the time of retreatment demonstrating response; 4 received cisplatin adjuvantly without measurable disease. 85% (11) of those patients are still alive. Of the 15/30 patients who ultimately achieved a complete remission, 11 received cisplatin during relapse treatment. 88% of the patients with cisplatin resistance (7/8) recognized during upfront therapy died from disease. Overall survival was associated with use of cisplatin at relapse (78.6% with vs. 25% without, *p* = 0.005), shown in Table 3. Average length of follow-up at time of analysis for surviving patients was 59 months. When excluding patients who were managed with palliative care, survival was again associated with cisplatin use at time of relapse (78.5% with cisplatin, 30.8% without, *p* = 0.021) (Figure 1). Examining only patients with upfront cisplatin sensitivity, survival was not significantly different between patients who received cisplatin and those who did not (70% vs. 37.5%, *p* = 0.34).

## 4. Discussion

Hepatoblastoma is the most common solid tumor in children and usually is cured with upfront therapy. Relapses occur in 10–15% of patients and refractory status occurs in 10%. There is no current standard of care for such patients and overall survival remains unacceptably low. Neither chemotherapy nor surgery alone is sufficient to manage most cases of relapsed hepatoblastoma, though in rare cases, such as limited and delayed lung relapse, surgery may prove sufficient as described by Shi et al. [12]. 

To date, there are only two publications describing experiences with relapsed hepatoblastoma in greater than 30 affected children, coming from the SIOPEL1-3 studies in Europe (59 patients) and the INT0098 study in the United States (36 patients) [3,5] respectively. The SIOPEL study highlighted the need for combined treatment with chemotherapy in conjunction with surgical removal of the tumor and described an average time to relapse of 12 months and three-year overall survival of 43% [1]. However, patients who failed to achieve a CR with upfront therapy were excluded from that report and thus the survival data is likely an overestimate of true survival for relapsed AND refractory hepatoblastoma patients treated on such studies. Nonetheless, factors associated with a favorable outcome were PRETEXT group I-III at diagnosis, a high AFP level at relapse, and a relapse strategy that included a combination of chemotherapy and surgical intervention. The INT0098 described a cohort of relapsed or refractory hepatoblastoma patients and reported that doxorubicin is an effective salvage chemotherapy at time of relapse or in high-risk disease that fails to respond to standard chemotherapy [3] in patients who did not receive doxorubicin as part of front-line treatment. Additionally, Zsiros et al. describes a cohort of 24 patients from the SIOPEL 1–4 studies in Europe who were treated with irinotecan at time of relapse or refractory status [13]. In this cohort, PFS at 1 year was 24% but irinotecan was well tolerated and was more effective in patients with isolated lung relapse and in those with initially chemosensitive disease compared those with initially chemorefractory disease. Aside from these three studies, only case studies with 1–3 patients each or comments in upfront trials with incomplete information are reported [14,15,16,17,18,19].

This retrospective analysis of experiences rrHB at a single institution aimed to evaluate the patterns of relapse, treatment options and overall outcome. This cohort of 30 patients showed similar time to relapse as previously published work with a slightly improved overall survival, despite inclusion of patients with initially refractory disease in addition to relapse patients [3]. In addition to chemotherapy, which at times re-introduced cisplatin-based therapy, many in the presented cohort also underwent aggressive surgical care including liver transplantation when appropriate, both factors of which may contribute to the improved overall survival despite the high-risk nature of the cohort. 

Understanding presenting features of relapsed HB is essential as it is a relatively rare entity. Standard of care off-therapy surveillance at our institution includes serial serum AFP monitoring along with risk-adapted radiographic imaging, the latter of which is controversial and not mandated on cooperative group trials of hepatoblastoma, noting that relapse with a normal AFP is uncommon. Similar to that reported in the SIOPEL series, we report several cases of relapse with a normal AFP (approx. 7%). While a normal AFP at time of relapse was associated with poor outcomes in the SIOPEL series, both patients in this cohort with normal AFP at time of relapse are alive and in remission [3]. This may relate to improved pathology analytics in the current era; where we were able to rule out rhabdoid tumor in our patients; the SIOPEL series may have included some such patients. Serial imaging is a not an entirely benign intervention due to radiation exposure for chest surveillance and anesthesia needs, as well as the anxiety it provokes in patients, parents and care providers. It is also not clear that imaging will have improved the outcomes of our 2 cases presenting with a normal AFP, as the AFP may have ultimately risen and signaled relapse. Further refinements of risk stratification of imaging surveillance and the advance of circulating tumor DNA as a liquid biopsy whereby cancer residue can be detected from blood sampling are worth further exploration [20]. With increased availability of next generational sequencing, further investigation should work to understand the genetic alterations that carry worse prognosis or higher likelihood of aggressive disease. 

Importantly, there were a half dozen patients in this cohort who had multiply relapsed disease. Presented data shows the same cure rate in these patients as compared to those with a single, isolated relapse. This suggests that multiply relapsed hepatoblastoma can still be cured with a combination of aggressive surgical approaches and adjuvant chemotherapy. Experience with this cohort suggests that patients with initial cisplatin resistance, however, tend to have less favorable histology and is associated with poorer prognosis. 

Limitations in this study include its retrospective nature including a several decade single institution experience with associated biases for care and a slightly smaller number of patients compared with cooperative group trials. There was variability between subjects regarding cisplatin dosing and schedule, a well-described cisplatin retreatment regimen will be needed to fully understand tolerability and efficacy. As a transplant center, there could be a referral bias with increased number of cases with liver relapses that would require transplantation for local control. Additionally, when considering surgical interventions, this cohort was primarily treated before the routine use of intraoperative indocyanine green (ICG) fluorescence guidance for excision at this institution [21], which is an opportunity for possibly improved surgical clearance. Toxicity data was limited, particularly for patients who received only parts of their care at this institution. Given the wide variety of regimens used, it is challenging to draw significant conclusions about toxicity. There was incomplete ototoxicity monitoring of all patients treated at relapse. 

Nonetheless, the presented cohort of relapsed/refractory hepatoblastoma patients demonstrates that retreatment with cisplatin as part of relapse therapy can be a therapeutic option that is feasible and such therapy may portend an improved overall survival. Larger prospective trials will be necessary to confirm such findings. Considerations for total toxicity and cumulative effect, particularly on hearing and renal function, remain at the forefront. There is insufficient data to understand the impact of salvage platinum exposure on further hearing loss in patients who have ototoxicity from upfront therapy. Further work to determine if otoprotective agents such as sodium thiosulfate can mitigate further toxicity while not impacting tumor response during salvage platinum therapy is essential. Efforts to improve cisplatin sensitivity may be one option to improve the outcome for patients with resistance, such as combination therapy with biological modifiers. Evidence for synergy with drugs inhibiting HDAC1 along with cisplatin have been reported in vivo in hepatoblastoma models as well as through connectivity mapping [22,23]. Given the established feasibility of combining such agents in pediatric cancer patients, this may present as one option for further clinical investigation in patients with cisplatin refractory disease.

## 5. Conclusions

Relapsed and refractory hepatoblastoma remains a challenge to treat with no general consensus on the most effective chemotherapy regimens. This report of a single institutional experience with relapsed/refractory HB demonstrates the clinical importance of cisplatin sensitivity and suggests an important role for re-treatment with cisplatin at time of relapse. 

## Figures and Tables

**Figure 1 cancers-15-03921-f001:**
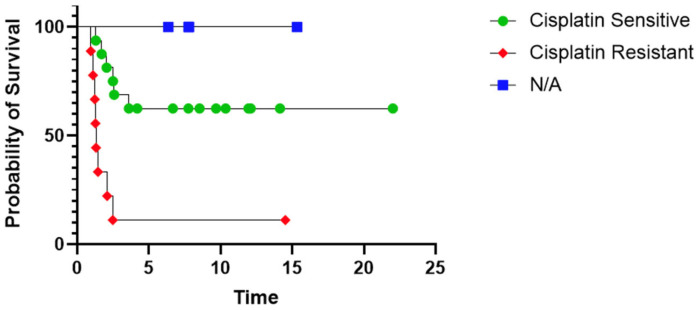
Patients with cisplatin sensitive disease during upfront therapy showed improved overall survival compared to those with cisplatin resistant disease.

**Table 1 cancers-15-03921-t001:** Demographics of the rrHB cohort.

DEMOGRAPHICS (*n* = 30)				
AGE		Median	Average	Range
	Initial	19.5 mo	28.8 mo	3–169 mo
	Relapse	32.5 mo	41.3 mo	12–194 mo
		#	%	
GENDER				
	Male	19	63%	
	Female	11	37%	
ETHNICITY				
	Caucasian	21	70%	
	Hispanic	4	13%	
	African American	2	7%	
	Middle East Asian	2	7%	
	Other	1	3%	
GESTATIONAL AGE				
	<37 weeks	4	13%	
	≥37 weeks	26	87%	
PREDISPOSITION SYNDROMES				
	Beckwith-Wiedemann	1	3%	
	Trisomy 18	1	3%	
	ARPKD/Caroli	1	3%	
PRETEXT at initial diagnosis				
	I	1	3.3%	
	II	10	33.3%	
	III	7	23.3%	
	IV	11	36.7%	
	Missing	1	3.3%	

**Table 2 cancers-15-03921-t002:** Cohort genetic alterations identified via next generation sequencing at time of upfront diagnosis.

Identified Tumor Mutations			
	#	%	
CTNNB1	13	76%	
NRF2	4	24%	CTNNB1 mutations were associated with: NRF2 (2), SMARCA4 (1) BCOR & TERT (1), RAD51 & PTPRO (1), MDM4 & NF1 (1), MLL3 (1), GSK3B & LRP1B (1), CREBBP & ADA (1)
MDM4	2	12%	
SMARCA4	1	6%	
BCOR & TERT	1	6%	
ARID1A	1	6%	
MUTYH	1	6%	
ATM E668	1	6%	
NF1	1	6%	
MLL3	1	6%	NRF2 mutations were associated with: CTNNB1 (2), ARDIA1 & MDM4 (1)
GSK3B & LRP1B	1	6%	
CREBBP & ADA	1	6%	
RAD51 & PTPRO	1	6%	
None	1	6%	

**Table 3 cancers-15-03921-t003:** Cisplatin utilization during relapse therapy in patients with initial cisplatin sensitivity and resistance during upfront therapy.

		Cisplatin Response during Upfront Therapy						
		Cisplatin Sensitive (*n* = 18)		Cisplatin Resistant (*n* = 8)		Cisplatin status N/A (*n* = 4)		Overall Survival
		Alive	Deceased	Alive	Deceased	Alive	Deceased	
Chemo Regimen during relapse	+ cisplatin	7	3	1	0	3	0	78.5%
	No cisplatin	3	5 *	0	7 **	1	0	25%
* 2 subjects were managed with palliative care and did not receive chemotherapy per family preference ** 1 subject was managed with palliative care and did not receive chemotherapy per family preference								*p* = 0.005

## Data Availability

The data presented in this study are available upon request from the corresponding author.

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
