# Peer review of "Retreatment with Cisplatin May Provide a Survival Advantage for Children with Relapsed/Refractory Hepatoblastoma: An Institutional Experience"

_cancers, 2023, doi:10.3390/cancers15153921_

Round 1

Reviewer 1 Report

Relapsed and refractory hepatoblastoma is one of the most challenging aspects in its treatment. The treatment outcomes are influenced by both surgical interventions and chemotherapy. Single-center studies hold significant value.

The study is rich in data subjected to analysis, and the methodology and results are described in detail. I am missing some data. What percentage of all your patients with hepatoblastoma have relapsed/refractory hepatoblastoma? What is the observation period for patients who have survived? What was the time between recurrence and death? I am particularly interested in patients with unresectable tumors who require liver transplantation qualification. In your study, there were 9 patients who underwent liver transplantation as the initial surgical intervention, and transplantation was also used as a treatment for recurrent cases in several patients. I am curious about the outcome of these transplants. Unfortunately, in our experience, liver transplantation outcomes for recurrent cases have been very poor.

Reviewer 2 Report

Retreatment with cisplatin may provide a survival advantage for children with relapsed/refractory hepatoblastoma: an institutional experience CANCERS 2023

Nice paper, little published on relapsed/ refractory HB

ABSTRACT

Results:

It states: Survival was associated with use of cisplatin at relapse (78.6% with vs 25% without, p = 0.012).

In conclusion: Retreatment with cisplatin at the time of relapse may provide an advantage for some patients with hepatoblastoma.

Question: how many patients were retreated and which dose (-schedule) had they received?

INTRODUCTION

First-line standards of care account for such various prognostic criteria resulting in risk-adapted therapy [7].

Question: what is meant with this?

to interval compressed cisplatin-based therapy including doxorubicin and other agents.

Question: please clarify interval compressed cisplatin therapy.

METHODS

An independent pathology review including immunohistochemical staining of all samples from time of initial diagnosis and relapse was performed by the CCHMC Department of Pathology for clinical purposes

Comment: this is is a big plus

A board-certified pediatric radiologist with more than 10 years of experience reviewed the imaging components of this study

Question: only one pediatric radiologist? No consensus reading? From the same institute (potential bias)?

Cisplatin chemosensitivity was defined as >1 log fold decrease in AFP and/or >30% reduction of tumor volume.

Question: where does this come from? Any references for this?

Question: Did the radiologist also look at pulm mets?

RESULTS

N=31

Comment: rather slow number, single centre experience. Could these numbers be expanded?

Subjects were treated with multiple chemotherapy regimens, including cisplatin, 5- floururacil, vincristine and doxorubicin per COG AHEP0731 protocol and cisplatin with doxorubicin per SIOPEL4 A1-3.

Question: perhaps reflect the Cisplatin on a scale of timeline and cumulative dose.

TABLE 3 is very helpful

Outcomes:

The overall survival for relapsed refractory patients was 50% (15/30 subjects), with one subject currently receiving ongoing chemotherapy. Subjects who died due to disease had a shorter average time between initial diagnosis and relapse as compared to those who survived (9.11 months vs 15.45 months, p = 0.06).

Comment: quite promising a OS of 50%, how many years OS?

Figure 1: cisplatin sensitive-/ resistant

Comment: very clear difference

Question: could you elaborate, what makes patients sensitive/ resistant? Biology? Genetics?

DISCUSSION

Further refinements of risk stratification of imaging surveillance and the advance of circulating tumor DNA as a liquid biopsy whereby cancer residue can be detected from blood sampling are worth further exploration

Question: would be liquid biopsy be of future use? Evidence for this?

Author Response

Responses added to the revised manuscript. We really appreciate your thoughtful questions. Some are frustrating limitations of a retrospective study with incomplete data availability, but we have attempted to address all questions. The root of cisplatin resistance remains an area of active thought and investigation within our group as we suspect it arises from a combination of biology and genetics and might hold the key to understanding why some hepatoblastomas are more aggressive from day 1.